# Progress on Respiratory Syncytial Virus Vaccine Development and Evaluation Methods

**DOI:** 10.3390/vaccines13030304

**Published:** 2025-03-12

**Authors:** Lie Deng, Hongjie Cao, Guichang Li, Kaiwen Zhou, Zihan Fu, Jiaying Zhong, Zhongfang Wang, Xiaoyun Yang

**Affiliations:** 1Guangzhou National Laboratory, Guangzhou 510320, China; 2State Key Laboratory of Respiratory Disease, National Clinical Research Center for Respiratory Disease, Guangzhou Institute of Respiratory Health, The First Affiliated Hospital, Guangzhou Medical University, Guangzhou 510180, China

**Keywords:** respiratory syncytial virus, vaccines, vaccine evaluation, immunological surrogates

## Abstract

Respiratory syncytial virus (RSV) remains a significant global health threat, especially to infants, the elderly, and immunocompromised individuals. This review comprehensively explores the progress in RSV vaccine development, the immune evaluation methods, and immunological surrogate. The RSV fusion (F) protein, a primary target for vaccine development, has been engineered in prefusion conformation to elicit potent neutralizing antibodies, while the attachment (G) glycoprotein and other immunogens are also being explored to broaden immune responses. Advances in diverse vaccine platforms, ranging from live attenuated and protein subunit vaccines to cutting-edge mRNA- and nanoparticle-based formulations, highlight the field’s progress, yet challenges in balancing safety, immunogenicity, and durability persist. Central to these efforts is the identification and validation of immunological surrogates, which may serve as critical benchmarks for vaccine efficacy. Neutralizing antibody titers, multifunctional T cell responses, and B cell memory have emerged as key correlates of protection. However, the feasibility of these surrogates depends on their ability to predict clinical outcomes across diverse populations and settings. While neutralizing antibodies block the virus directly, T cell responses are essential for clearing infected cells and preventing severe disease, and B cell memory ensures long-term immunity. Integrating these immunological markers into a cohesive framework requires standardized assays, robust clinical validation, and an in-depth understanding of RSV-induced immune response.

## 1. Introduction

Respiratory syncytial virus (RSV) is one of the primary pathogens that cause respiratory tract infections in infants, young children, and older adults, often causing pneumonia or bronchitis. The RSV infection pattern exhibits a U-shaped age curve, with the highest incidence in individuals aged < 5 years and >60 years [1]. Globally, approximately 30 million children under 5 years of age are infected with RSV annually. In 2019, there were 33.0 million episodes of RSV-associated acute lower respiratory infections, leading to approximately 3.6 million hospitalizations, 26,300 in-hospital deaths due to RSV-associated acute lower respiratory infections, and 101,400 overall deaths attributable to RSV in children aged 0–60 months. [2]. Findings from the Pneumonia Etiology Study in Childhood Health showed that RSV is associated with pneumonia in children under 5 years of age [3]. RSV is an important cause of illness in community-dwelling older adults. In the United States alone, approximately 177,000 adults over 65 years of age are hospitalized annually for respiratory tract infections caused by RSV, with 14,000 fatalities recorded [4]. RSV can result in more severe illnesses, such as pneumonia, in older adults, and worsen respiratory conditions, including asthma and chronic obstructive pulmonary disease. The significant health burden of RSV has made vaccine development a priority for more than five decades. However, to date, vaccine protection for key populations, including infants less than 6 months old, young children 6–24 months old, pregnant women, and people over 65 years old, remains limited.

Formalin-inactivated RSV (FI-RSV) was the first RSV vaccine that entered clinical trials. Unfortunately, it failed, as recipients developed enhanced respiratory disease (ERD) upon exposure to natural RSV infection [5]. It was reported that FI-RSV could not induce the production of neutralizing antibodies or CD8+ T cells, but it elicited cytokine responses, leading to ERD [6]. This adverse outcome was attributed to the fact that the virions of FI-RSV predominantly contained the post-fusion form (post-F) of the F protein while lacking the pre-fusion form (pre-F), which is known to induce neutralizing antibodies [7]. So far, although more than 30 candidate vaccines have entered clinical trials or are in preclinical developmental stages, only 3 RSV vaccines have been approved by the Food and Drug Administration (FDA). Overall, RSV vaccine development still faces challenges, including (i) the occurrence of ERD in vaccinated individuals following RSV infection, (ii) problems associated with inducing and evaluating protective immunity, (iii) the high costs of vaccine clinical trials, (iv) the infection of infants and young children with RSV, increasing the complexity of vaccine development and deployment, and (v) the establishment of long-term protective immunity. Currently, multiple RSV vaccine development platforms are being utilized, including live attenuated subunit, viral vector, nanoparticle vector, and mRNA vaccines. In this review, we briefly summarize the main findings in the field of RSV vaccine research and development in recent years, including the progress of various vaccine developments, evaluation of RSV vaccines, and the prospects of RSV research and clinical intervention.

## 2. RSV Structure and Vaccine Targets

### 2.1. Virus Structure

RSV is a single-stranded, negative-sense, non-segmented RNA virus with a genome length of approximately 15 kb, encoding 11 structural and non-structural proteins, including externally exposed transmembrane glycoproteins (small hydrophobic protein SH, adhesion protein G, and fusion protein F), internal structural proteins (matrix protein M and nucleoprotein N), functional polymerase complex proteins (phosphoprotein P and polymerase L), non-structural proteins (NS1 and NS2), and regulatory M2 proteins (M2-1 and M2-2) (Figure 1). The life cycle of RSV starts with attachment to host cells via F and G proteins. Subsequently, viral–host membrane fusion releases the genome. Genome-templated transcription and replication occur, followed by the assembly of new virions. Finally, virions are released by budding (Figure 2). The 11 proteins play orchestral roles in the life cycle. RSV G protein binds to chemokine (C-X3-C motif) receptor 1 (CX3CR1) and Toll-like receptor 4 (TLR4) on the cell surface and initiate the its attachment. The F protein binds to cellular TLR4 and nucleolin and mediates the fusion and entry of the nucleocapsid into the cytoplasm, and it also plays a crucial and multifaceted role in virus cell–cell transmission and the formation of syncytia. The non-structural proteins NS1 and NS2 play a crucial role in evading the host’s innate immune defenses. Specifically, NS1, with its distinct structural domain, is instrumental in modulating host immune reactions, such as suppressing type I interferon (IFN) responses, hindering the maturation of dendritic cells, and enhancing inflammatory processes [8]. On the other hand, NS2 suppresses the ubiquitination of dormant retinoic acid-induced gene I and melanoma differentiation-associated protein 5, thereby blocking subsequent signaling pathways and IFN synthesis [8]. Phosphoproteins (P) function as vital polymerase cofactors. They serve as a cofactor for the N protein monomer, linking the L protein to the nucleoprotein–RNA (N-RNA) assembly [9]. Moreover, they act as molecular chaperones that inhibit the newly produced N protein from attaching to the cellular RNA [10]. The small hydrophobic (SH) protein is a pentameric ion channel believed to be associated with delayed apoptosis of infected cells [11]. The M protein, residing within the viral envelope, structurally supports the envelope and participates in the transcription of viral RNA.

### 2.2. Target Immunogens of the RSV Vaccine

The 11 proteins in the RSV structure play important roles in its life cycle and represent potential targets for RSV vaccine development. Currently, more than 30 candidate vaccines are under clinical or preclinical development, mainly targeting the F, G, N, and SH proteins. 

The F protein serves as the primary target for RSV vaccine development. It is the most abundant enveloped protein and is highly conserved across various viral strains, with only 25 amino acid variations distinguishing RSV subtypes A and B [12]. The F protein plays a dual role in adhesion and fusion, enabling it to interact with glycosaminoglycans and nucleolin present on the cell surface. This interaction facilitates viral entry into cells and promotes the formation of syncytia. As a class I fusion protein, the F protein consists of 574 amino acids (AA) and features two furin cleavage sites located at AA positions 109 and 136. After furin cleavage, a C-terminal F1 subunit (50 kDa), an N-terminal F2 subunit (20 kDa), and a 27 AA fragment are generated. Initially, F1 and F2 form dimers, which subsequently assemble into trimeric pre-F structures [13]. During viral entry, the pre-F conformation undergoes a dramatic structural rearrangement, transitioning to the post-F state, facilitating the fusion of viral and host cell membranes. The pre-F conformation is metastable and susceptible to transforming into the post-F state. The majority of neutralizing epitopes are located on the pre-F protein rather than the post-F protein. The pre-F protein possesses six antibody epitopes, namely, sites I, II, III, IV, V, and Ø (Figure 3). In contrast, the post-F protein contains only four epitopes, namely, sites I, II, III, and IV, lacking the V and Ø sites, which are recognized as potent neutralizing epitopes. Therefore, stabilizing the F protein in its pre-F conformation is of fundamental importance in the design and development of RSV vaccines. In recent years, the research and development (R&D) of RSV vaccines have predominantly centered on the pre-F protein as the core target. Structure-based rational antigen design, including DS-Cav1 [14], SC-TM [15], sc9-10 DS-Cav1 [16], and 847A [17], etc., have made considerable achievements in stabilizing the pre-F protein, and multiple preclinical and clinical studies demonstrate promising immunogenicity and safety profiles of pre-F-based vaccine candidates. Three approved RSV vaccines are all pre-F-based. The first two approved RSV vaccines in the United States are protein-subunit vaccines based on the stabilized pre-F conformation. GSK’s RSVPreF3 vaccine incorporates a liposome-based adjuvant to boost immune response, while Pfizer’s RSVPreF vaccine is formulated without an adjuvant [18,19]. mRNA-1345, manufactured by Moderna, is an RSV pre-F form coupled with lipid nanoparticles (LNPs) [20]. 

The G protein is a surface glycoprotein that mediates RSV attaching to host cells. It exhibits two forms, membrane-bound (mG) and secreted (sG), of which the mG form mediates viral attachment. The mG protein, a type II membrane protein, is heavily glycosylated with 30–40 O-linked and 3–5 N-linked glycans, featuring an extracellular domain composed of an unglycosylated central conserved region flanked by two unstructured, highly variable mucin-like domains. The central conserved region, which contains 13 strictly conserved amino acids across all RSV strains, plays a critical role in eliciting a protective immune response [21]. The surface residues of these AAs overlap with the cystine lasso containing a 1-4, 2-3 disulfide bond topology, which is considered to mediate the attachment to human airway epithelial cells via interaction with CX3CR1 during natural infection [22]. Most neutralizing antigenic epitopes are located on mG proteins, making them potential targets for vaccine development. Despite the demonstrated efficacy of G-protein-targeted vaccines in eliminating RSV within animal models [23,24], to date, such vaccines have not yet received approval. Furthermore, considering the different inhibitory phases of infection associated with different vaccine types, there is an anticipation that G-targeted vaccines, when combined with F-targeted vaccines, may yield enhanced effectiveness [25]. This potential synergy could offer a more comprehensive approach to combat RSV infections, highlighting the need for further research and development in this area. 

Nucleoprotein (N), together with phosphoprotein (P), covers the RSV RNA genome within the nucleocapsid and protects it from degradation. Besides its key structural role, the N protein plays a role in immune evasion. Additionally, the N protein, as the most conserved RSV protein with high structural constraints, is a major target of the CD8+ T cell response induced by natural infection. Thus, a vaccine formulation combining the N protein and F-neutralizing epitopes has the potential to induce a broad spectrum of cross-protective immunity against divergent RSV strains and minimize the emergence of escape mutants. BCG-N RSV is the first N-protein-targeting vaccine to enter clinical trials [26]. Another poxvirus-based RSV vaccine in a phase II clinical trial targets several proteins, including the N protein [27].

The SH protein, a type II protein comprising 64–65 AAs on the viral surface, forms a pentameric cation-selective ion channel (viroporin) and activates the NLRP3 inflammasome, promoting IL-1β expression [28]. Live attenuated RSV vaccine candidates often involve deletion of the SH gene. Although SH does not induce a neutralizing response [28], SH vaccines induce antibodies that mediate ADCC and exhibit protective effects [29]. Therefore, the SH protein can be used as a target for RSV vaccine design. A vaccine based on the extracellular domain of the small hydrophobic glycoprotein (SHe) protected mice and cotton rats against intranasal RSV challenge, significantly reducing pulmonary viral replication [29]. In a phase I study, the SH-targeting vaccine also demonstrated immunogenicity in adults aged 50–64 years [30].

## 3. Progress in RSV Vaccine Research and Development

As early as the 1960s, several clinical studies on inactivated vaccines had been carried out [31]. However, the formalin-inactivated respiratory syncytial virus (FI-RSV) vaccine not only failed to provide protection but also led to a vaccine-associated enhanced respiratory disease (VAERD), causing vaccinated children to experience more severe symptoms after RSV infection and unfortunately resulting in two deaths [32]. This catastrophic outcome decelerated the R&D of RSV vaccines targetsing F for decades, until the decoding of the dominance of the post-F protein on FI-RSV virions. Since then, the discovery of importance of the stabilized pre-F protein in inducing neutralizing antibodies revitalized RSV vaccine R&D. Currently, three RSV vaccines have been approved: The Arexvy vaccine developed by GSK was first approved in the United States on 3 May 2023 for the prevention of lower respiratory tract infections caused by RSV in older adults over 60 years of age. The Abrysvo vaccine developed by Pfizer was first approved by the FDA on 31 May 2023. It is used for maternal immunization and is aimed at protecting infants and preventing RSV-induced lower respiratory tract infections in older adults [33]. mRESVIA (mRNA-1345) developed by Moderna was approved by the FDA to protect the elderly (≥ 60) from lower respiratory tract disease caused by RSV infection on 31 May 2024 [34]. Additionally, a number of vaccines are currently in the preclinical research stage or clinical trial phase (Table 1). Various vaccine platforms currently targeting RSV include live attenuated vaccines, subunit vaccines, vectored vaccines, nanoparticle vaccines, and mRNA vaccines, and the progress and developmental stage of these are summarized in the following section.

### 3.1. RSV Live Attenuated Vaccine

In the 1980s and 1990s, with the in-depth understanding of the biological characteristics of the RSV and the development of vaccine technology, researchers began to attempt the development of live attenuated RSV vaccines. Natural RSV infection induces robust humoral and cellular immune responses, making live attenuated vaccines an ideal strategy, particularly in the early stages of vaccine development. Live attenuated vaccines are pathogens that have undergone mutations and the deletion of pathogenic genes, and they retain antigenic proteins that induce host immunity, thereby weakening the virus’s virulence while maintaining its immunogenicity. They may provide long-term or lifelong protection while preventing the occurrence of the disease. Infants and young children vaccinated with live attenuated RSV vaccines do not experience aggravation of their condition when re-infected with RSV, indicating that these vaccines are suitable immunization strategies for this population group [35]. Previously, live attenuated RSV vaccines were prepared by changing the culture environment of the virus to cause adaptive mutations; however, this method is time-consuming and difficult to predict. Over the years, advancement in live attenuated vaccine research has used reverse genetics technology to analyze viral genes, knock out or mutate one or more viral functional proteins, and weaken viral replication ability while retaining immunogenicity. The advantage of live attenuated vaccine technology is its ability to mimic the natural infection process, stimulating an immune response without causing aggravation of respiratory diseases [36]. One design strategy for the generation of live attenuated RSV vaccines is to delete the RSV NS2 protein, an antagonist of the host interferon. The most advanced NS2-deficient live attenuated RSV vaccine is RSV/ΔNS2/Δ1313/I1314L, which exhibits low toxicity and moderate temperature sensitivity [37]. A second live attenuated RSV vaccine was designed by deleting most of the open reading frame encoding the M2-2 protein. Loss of M2-2 leads to higher viral gene transcription and reduced genome replication [38]. However, owing to the unpredicted potential of reverse mutations, live attenuated vaccines are considered the less-preferred RSV vaccine option. Even though a lot of effort has been made, no live attenuated vaccines have been approved yet.

### 3.2. RSV Protein Subunit Vaccine

Subunit vaccines are a type of vaccine purified from specific antigenic protein fragments of pathogens. They contain only key antigenic components capable of triggering an immune response, thus avoiding side effects potentially caused by non-essential pathogen components. This design endows them with high safety, making them suitable for the vaccination of a wide range of people, including those with weakened immunity or special health conditions. However, subunit vaccines are generally relatively less immunogenic. As a result, they often require the addition of adjuvants to enhance the ability to stimulate a robust immune response [39]. The main targets of RSV subunit vaccine design are the three membrane glycoproteins: the F, G, and SH proteins. Currently, two subunit RSV vaccines targeting pre-F proteins, RSVPreF and RSVPreF3, have been approved [33,40]. A phase III clinical trial conducted with the adjuvanted RSVPreF3 vaccine (AReSVi-006) reported an 82.6% efficacy against RSV-LRTD over a median follow-up of 6.7 months [18]. Similarly, a phase III trial conducted with RSVPreF in pregnant women (24th–36th weeks of gestation) showed an 81.8% efficacy against severe RSV-LRTD and a 57.1% efficacy against RSV-LRTD of any severity within 90 days post-birth [41]. FG-Gb1 is an RSV subunit vaccine for intranasal vaccination, produced by genetically fusing the core fragments of the F and G proteins of RSV with Gb-1 (a microfold cell-specific ligand) using a flexible linker. Mice vaccinated with FG-Gb1 had a significant increase in antigen-specific serum IgG, IgA, and neutralizing antibody titers, and FG-Gb1 vaccination effectively protected mice from RSV infection [42]. 

### 3.3. Vectored Vaccines

Vectored vaccines use viruses or bacteria that are nontoxic or less toxic to humans as vectors to express vaccine antigens in cells to induce an immune response. Viral and bacterial vector vaccines can simulate the natural infection process and elicit both innate and adaptive immune responses. However, the immunogenic effects of vectored vaccines may be compromised by pre-existing vector immunity. RSV vector vaccines currently under development include vectors such as BCG bacteria, Sendai virus, vesicular stomatitis virus, parainfluenza virus, adenovirus, and vaccinia virus, among which several have entered the clinical trial stage [26,43,44,45,46,47,48]. BLB201 is a live viral vectored RSV vaccine based on parainfluenza virus type 5 (PIV5) encoding the RSV F protein [45]. PIV5 has the advantages of safety and stability as a vaccine vector and exhibits weak pathogenicity, and simple PIV5 infection generally does not cause clinical symptoms. Notably, no vectored vaccine against RSV has obtained approval up until now.

### 3.4. mRNA Vaccines

mRNA vaccines mark a revolutionary step in vaccinology. Their rapid development, crucial for emerging diseases, and high immunogenicity, which effectively primes the immune response, are major advantages [49,50]. Nevertheless, mRNA vaccines confront several challenges. Their inherent instability, as mRNA molecules are prone to degradation, complicates their storage and transportation. Additionally, since naked mRNA has difficulty entering cells, the development of a safe and efficient delivery system is imperative. This remains a key research focus for unlocking the full potential of mRNA vaccines [49,50]. However, a major breakthrough was recently achieved in the development of mRNA targeting RSV. Moderna mRNA-1345 has been approved by the FDA for the prevention of RSV infections in individuals over 60 years of age. The vaccine exhibited an 83.7% efficacy against RSV-LRTD with at least two signs or symptoms and 82.4% against the disease with at least three signs or symptoms [51]. Several mRNA vaccines against RSV infections have been studied in clinical trials. Phase I clinical trials of mRNA-1777 have shown that it can promote the production of strong neutralizing antibodies against RSV [52]. mRNA vaccine candidates encoding stabilized pre-F or native F protein induced strong neutralizing antibody responses in mice and cotton rats, comparable to those elicited by an adjuvanted stabilized pre-F protein at equivalent doses. [53]. Recently, it was reported that an mRNA vaccine, based on a modified pre-F protein coupled with LNP (LC2DM-LNP), was designed to anchor to the cell surface and elicited high levels of neutralizing antibodies in both young and older female mice [54].

### 3.5. Nanoparticle Vaccines

Nanoparticle vaccines deliver specific antigens loaded onto nanocarriers. The chemical properties and small size of nanocarriers make nanoparticle vaccine platforms effective alternatives to traditional vaccines. Nanocarriers can be customized by changing the manufacturing process or material composition, which can alter the presentation of antigens and enhance their stability and conformation [55]. Beyond their structural versatility, nanoparticles possess intrinsic immunomodulatory capabilities. Many polymer-based and inorganic nanoparticles (such as polystyrene, titanium dioxide, and silica) have been shown to activate inflammasomes (e.g., the NLRP3 inflammasome), thereby triggering downstream immune responses and the production of cytokines (such as IL-1β, IL-6 and IL-18) [56,57,58]. Additionally, nanoparticles can achieve lung-specific deposition, particularly in the distal alveolar regions, by adjusting their size (e.g., 50–200 nm) and surface modifications, thereby directly targeting the initial sites of respiratory infections. For instance, intranasal administration of chitosan nanoparticles conjugated with cholera toxin significantly enhances mucosal IgA production [59], highlighting their potential to elicit robust protective immune responses at mucosal surfaces. Currently, several RSV candidate vaccines in preclinical testing or clinical trials use various nanoparticle-based carriers such as pre-F-NP [60] and ResVax [61]. However, no nanoparticle-based RSV vaccines have been approved yet.

**Table 1 vaccines-13-00304-t001:** Summary of RSV vaccine research and develsopment discussed in this review.

Platform	Examples	Target	Manufacturer	Status	References
Live attenuated vaccine	RSV/ΔNS2/Δ1313/I1314L	Deletion of NS2, deletion S1313 of L protein, I1314L of L protein of RSV	NA	Clinical phase II	[37]
	RSV 276	Deletion of M2-2 protein of RSV	NA	Clinical phase II	[38]
Subunit vaccine	RSVpreF3 (Arexvy)	Pre-F	GSK	Approved by FDA	[18,33]
	RSVpreF (Abrysvo)	Pre-F	Pfizer	Approved by FDA	[33,41]
	FG-Gb1	F and G protein	NA	Preclinical	[42]
Vectored vaccines	rBCG-N-hRSV	BCG vaccine expressing the nucleoprotein of RSV	IDT Biologika	Clinical phase I	[26]
	SeVRSV	Sendai virus expressing F protein	NA	Clinical phase I	[43]
	rVSV-G-2A-F	VSV expressing both the G and F proteins	NA	Preclinical	[44]
	BLB201	Type 5 (PIV5) that encodes the F protein	Blue Lake Biotechnology	Clinical phase I/IIa	[45]
	Ad26.RSV.preF	Adenovirus vector vaccine expressing pre-F	J&J/Jassen	Clinical phase IIb	[46]
	MVA-RSV	MVA expressing F, G, N, M2-1 protein of RSV A and G of RSV B	Bavarian Nordic	Clinical phase III (failed)	[48]
mRNA vaccines	mRNA-1345 (mRESVIA)	Pre-F	Moderna	Approved by FDA	[34]
	mRNA-1777	Pre-F	Moderna	Clinical phase I	[52]
	LC2DM-LNP	Pre-F	NA	Preclinical	[54]
Nanoparticle vaccines	ResVax	Pre-F	Novavax	Clinical phase III (failed)	[61]
	pre-F-NP	Pre-F	NA	Preclinical	[60]

NA: not applicable.

## 4. RSV Vaccine Testing and Evaluation

Safety and effectiveness are crucial concerns in vaccine R&D. Vaccine safety necessitates a thorough assessment of the potential risks and adverse effects caused by vaccine administration. Prior to embarking on human trials, vaccines need to be tested for safety and efficacy in preclinical testing, which encompasses laboratory and animal studies. This is followed by various phases of clinical testing. During these trials, the safety, immunogenicity, and efficacy of vaccines are meticulously examined. During clinical trials, any vaccine-associated adverse or side effects are closely monitored; however, this is not discussed in this review. Currently, the effectiveness of RSV vaccines is primarily determined based on their real-world protective efficacy. Previously, RSV vaccines were approved based on their efficacy over a follow-up period ranging from 90 days to 6.7 months [18,41,51]. However, building upon the experience of evaluating the immunogenicity of the COVID-19 vaccine, the immunological surrogate method is feasible for RSV vaccine evaluation with obvious advantages. This section focuses on the evaluation of vaccine effectiveness and immune substitution endpoint evaluation methods. 

### 4.1. Immunological Surrogates for Vaccine Protection and Their Evaluation Methods 

Although the effectiveness of vaccines can be evaluated by their protective effects in animals, animal models cannot completely predict their immunogenicity and effectiveness in humans. Typically, vaccines undergo phase I, II, and III clinical trials before being marketed. Phase III clinical trials evaluate vaccine effectiveness in large-scale target populations. Currently, the indicators of vaccine effectiveness include the infection prevention rate, severe disease prevention rate, and reduction in infection symptoms. The current method for the evaluation of RSV vaccine effectiveness focuses on the prevention of severe RSV-LRTD with at least two signs or symptoms and RSV-LRTD with at least three signs or symptoms. 

Verification of endpoint efficacy is critical for vaccine approval and marketing. However, not all vaccines qualify for efficacy endpoint verification. Verification of effectiveness usually requires time and infection coverage for at least one epidemic season. When evaluating two effective vaccines (positive control trials) or evaluating the incidence of clinical endpoint events, assessing the efficacy of the trial vaccine by observing clinical endpoint efficacy requires a larger sample size and an extended follow-up period, which may not be feasible in real-world settings. To address these issues, vaccine researchers have proposed the concept of immunological surrogate endpoints [62] and a method of immunogenicity-bridging clinical trials [63]. Immunological surrogate endpoints are important for the timeliness of vaccination against new or emerging viral infectious diseases. Compared to large-scale phase III clinical trials, immunological surrogate evaluation can save more than 60% in terms of time and over 80% in terms of expenses [64] and is thus recommended as a vaccine efficacy endpoint. For instance, a hemagglutination inhibition titer (HI) ≥  40 is recommended to be an immune surrogate for 50% protection to evaluate the efficacy of new candidates of influenza vaccines [65].

Immunological surrogate endpoints provide a time- and cost-efficient alternative to traditional clinical trials, enabling the rapid evaluation of vaccine efficacy, particularly for emerging infectious diseases. The World Health Organization (WHO) has recommended some immunological surrogate endpoints for vaccine-preventable diseases; however, research on surrogate endpoints for RSV vaccines is lacking. Based on the previous significant positive correlation between the immunogenicity and efficacy of RSV vaccines [66], neutralizing antibodies and cellular immunity can be used as surrogate endpoints for RSV vaccine evaluation.

### 4.2. Immunological Evaluation of RSV and Its Correlation with Protective Immunity 

Vaccination triggers a coordinated humoral and cesllular immune response. Humoral immunity, driven by B cells, involves antigen recognition, subsequent activation, and differentiation into antibody-secreting plasma cells and memory B cells, which facilitate rapid recall responses. Simultaneously, cellular immunity is orchestrated through the activation of T cells; CD4+ T cells differentiate into helper subsets that modulate immune activity via cytokine secretion, while CD8+ T cells transform into cytotoxic effectors capable of eliminating infected cells. B cells, aided by T cell co-stimulation, further amplify the response by generating additional plasma and memory cells. Therefore, the important aspects of the immune response consist of humoral, especially neutralizing, antibodies, T cell response and T cell memory, and B cell response and B cell memory. These integrating immunological markers comprise the immune profile after vaccination, and different methods have been developed for quantitative measurements (Figure 4). 

Neutralizing antibody (NAb) titer is commonly used to evaluate humoral immunity and is highly correlated with the protective effect and durability of RSV vaccines. Previous research has illuminated the significant role of RSV-related antibodies in protection against RSV-associated diseases. Two studies [67,68] have shown that in children under 6 months old, the levels of RSV neutralizing antibodies (NAbs) in cord blood are associated with protection from RSV-induced hospitalization. Concurrently, other research [69,70] has indicated a negative correlation between RSV-specific antibody levels and the severity of RSV disease and pneumonia. Findings from studies on cotton rats also support the protective effect of NAbs. For instance, when challenged intranasally, cotton rats were protected with a serum-neutralizing titer of approximately 1:380 [71]. 

Currently, various RSV candidate vaccines are under clinical development, many of which are designed to induce strong virus-neutralizing immune responses. Cellular immunity is essential for the production of NAbs and holds a pivotal position in safeguarding against RSV infection. Even though T cells are unable to prevent the initial infection, they prove to be instrumental in numerous infectious diseases. They can effectively reduce the viral load, mitigate clinical symptoms, and decrease the rates of severe cases and mortality. Based on the association between the cytotoxic CD8+ T lymphocyte (CTL) response and reduced disease severity [72], cellular immune responses are thought to contribute to viral clearance. Specifically, T-helper 1 (Th1) CD4+ T cells significantly influence the control of RSV infection. They achieve this by promoting the development of CTLs and enhancing the humoral response [73]. Moreover, the AS01-adjuvanted RSVPreF3 has been shown to trigger polyfunctional CD4+ T cell responses in mice, providing a strong basis for evaluating vaccine candidates based on AS01-adjuvanted RSVPreF3 in subsequent clinical trials [74]. 

Memory B cells, although not well recognized, are a preservative indicator of vaccine efficacy. They function with specific T cells to maintain antibody levels in the body and are highly correlated with antibody duration. However, no comprehensively determined correlation for predicting vaccine efficacy has been established so far. McGuire et al. reported the prediction of vaccine efficacy through reconstituted interactions between the antigen and germline B cell receptor (BCR) [75,76], and they suggested a new approach for the evaluation of vaccine candidates. Biochemical reconstruction of germline BCR stimulation, which mimics the natural process of B cell stimulation using biochemical means, is emerging as a crucial tool in vaccine design. It has been utilized to screen immunogenic candidates for vaccines against various pathogens, such as human immunodeficiency virus (HIV), H1N1, and H5N1 [75,76,77,78].

#### 4.2.1. Neutralizing Antibody Evaluation

Neutralization assays are determined by counting viral plaques or foci visualized by staining cells with crystal violet or immunostaining with antibodies or by the appearance of a cytopathic effect (CPE) in the wells upon exposure to differently diluted antibody samples [79]. In the realm of RSV research, the microneutralization test (MCPENT), which observes cytopathic effects (CPEs), and the plaque reduction neutralization test (PRNT), which counts plaque numbers, are the most commonly employed live-virus-based neutralization tests (Table 2). These methods have been utilized in the majority of preclinical and clinical trials to evaluate immunogenicity. However, to expedite vaccine development in animal studies and large-scale clinical trials, high-throughput methods, including the use of RSV viruses expressing green fluorescent protein (GFP) or a luciferase protein, have emerged [80,81]. When using a GFP-expressing virus, images can be automatically acquired, and the focus can be enumerated, enabling the screening of up to several hundred samples per day. Standardizing the RSV neutralization assay is crucial for accurately evaluating neutralizing antibody (NAb) responses to vaccine candidates. The WHO International Standard for Antiserum to RSV (RSV IS) enables consistent NAb titer measurements across laboratories and assay formats, supporting the comparison of immunogenicity and advancing vaccine development.

#### 4.2.2. T Cell Response Evaluation

T cell response assays are typically designed to measure immune responses following natural infection or vaccination. Enzyme-linked immunospot (ELISpot) assays, intracellular cytokine staining (ICS), and surface activation marker (AIM) analysis via flow cytometry are widely used to monitor antigen-specific T cell responses. ELISpot is the most convenient method, as it directly enumerates spots post-stimulation without requiring advanced equipment or specialized training. In contrast, ICS and AIM provide deeper insights by linking cytokines or activation markers to specific T cell phenotypes, enabling detailed functional characterization. While tetramers offer precise epitope-specific T cell detection, their reliance on specific HLA haplotypes limits their utility for population-level T cell response assessments; therefore, they are not used for vaccine evaluation. RSV-specific T cell responses are usually determined after stimulation with a pool of 15–20 mer overlapping peptides, covering the RSV-F protein and/or proteins of interest. The cytokines IFN-γ, TNF-α, IL-2, and IL-4 are frequently measured to determine RSV-specific T cell responses. ELISpot is often used in clinical trials for the evaluation of RSV vaccines. For instance, in a phase I clinical trial of the recombinant BCG vaccine expressing the nucleoprotein of RSV, multivalent poxvirus-based RSV vaccine in a phase II clinical trial, and an RSV pre-F protein vaccine (Ad26.RSV.preF) in a phase IIb clinical trial, cellular responses were assessed by ELISpot via the secretion of IFN-γ, TNF-α, IL-2, and IL-4 in peripheral blood mononuclear cells upon restimulation of the RSV protein peptide pools (F, G, M2, N) or RSV virion [26,27,82]. In preclinical trials of adenoviral vector vaccine candidates, RSV.FA2, AS01-adjuvanted RSV pre-F (RSVPreF3), and CD4+ and CD8+ T cell responses were determined using ICS [83,84]. 

#### 4.2.3. B Cell Memory Evaluation

B cells, particularly pathogen-specific memory B cells, represent the body’s humoral immune reserve for immunogens. Therefore, the determination of memory B cells may also be an important aspect of RSV vaccine immunology evaluation. This method is still relatively new and has only been evaluated for COVID-19, HIV, and influenza vaccines [75,78,85,86]. This method uses single-cell BCR technology, including the LIBRA-seq technique or nested BCR PCR followed by single-cell sorting to obtain the variable, diversity, and joining sequences of immunogen-specific B cells and determine the BCR repertoire, immunoglobulin heavy chain variable region usage, and CDR3 sequences. By comparing the findings with known BCR usage or existing high-affinity antibody sequences, a high-quality humoral response can be quantified. Moreover, the differences in the antibody components produced by different vaccines or immune doses can be compared. Schneikart et al. analyzed antigen-specific BCR repertoires of individuals vaccinated with post-F vaccines in a phase I trial and revealed a preferential expansion of VH4-encoded BCRs in response to vaccination. A comparison of the vaccine-induced BCR repertoires with those of the healthy donors suggested an expanded level of pre-/post-F cross-reactive B cell clones induced by vaccination [87]. Using CDR3 data and an antibody database, the antigen-specific antibody sequence (QASAS) method allows for the quantification of memory B cells to evaluate immunogenicity following COVID-19 mRNA vaccination [86,88]. The quantification of antigen-specific antibody sequences through BCR analysis represents a promising novel strategy for RSV vaccine evaluation. 

## 5. Conclusions

The development of RSV vaccines has been a significant area of research for over half a century. Despite the approval of several RSV vaccines, including two subunit vaccines and one mRNA vaccine by FDA, the fight against RSV is far from over. These approvals highlight progress but also the ongoing challenges in effectively protecting vulnerable groups like infants, young children, and older adults. Each vaccine platform has its own pros and cons. Live attenuated vaccines are relatively easy to develop but may have a small risk of reversion. Subunit vaccines are relatively safer due to their purity but may need adjuvants for better immunogenicity. Vector-based vaccines can deliver antigens effectively but are affected by pre-existing vector immunity. mRNA vaccines offer rapid development and high immunogenicity but face stability and delivery system issues. Nanoparticle vaccine platforms can enhance immune response but have production and cost challenges. So far, three RSV vaccines, based on the subunit protein and mRNA platform, have currently been approved, whereas attenuated vaccine, vectored vaccine, and nanoparticle vaccine platforms remain unapproved. 

The majority of neutralizing epitopes are located on the pre-F rather than the post-F protein, but the pre-F protein is vulnerable to transitioning to the post-F protein. Thus, stabilizing the pre-F conformation is crucial to vaccine development. Structure-based-designs, such as Cav1, SC-TM, sc9-10 DS-Cav1, and 847A, for pre-F stabilization, have made substantial progress in RSV vaccine development. Stabilizing pre-F ensures the proper presentation of immunogenic regions to the immune system, maximizing the vaccine’s protective capacity, thereby enhancing the vaccine’s effectiveness and potentially broadening its application to more population groups. All the approved vaccines, RSVpreF3 (GSK), RSVpreF (Pfizer), and mRNA-1345 (Moderna), have utilized stabilized pre-F as targets. In summary, pre-F is fundamental for successful RSV vaccine development, and continuous research in this area is vital for the better prevention and control of RSV-related diseases. 

Conducting clinical trials, especially large phase III trials in vulnerable populations, involves strict approval processes, slowing development and emphasizing the need for reliable evaluation methods. Besides real-world efficacy, immunological surrogates are a viable alternative, and measuring vaccine-induced immune markers is crucial for correlating RSV vaccines with protection. A variety of serological and cellular immunoassays have been developed. Serological methods like MCPENT or PRNT for NAb detection and cellular assays like AIM or ELISpot for T cell response evaluation, along with B cell responses, are important. However, comparing immune responses of different vaccine candidates is difficult due to a lack of standardization. The establishment of international standards for immune marker assessment and international multicenter clinical trial collaboration are essential for accurately determining vaccine protection correlations. In the future, RSV vaccine development is likely to trend towards platforms that combine high immunogenicity, safety, and the ability to rapidly adapt to viral variants, such as optimized mRNA or nanoparticle-based approaches. 

## Figures and Tables

**Figure 1 vaccines-13-00304-f001:**
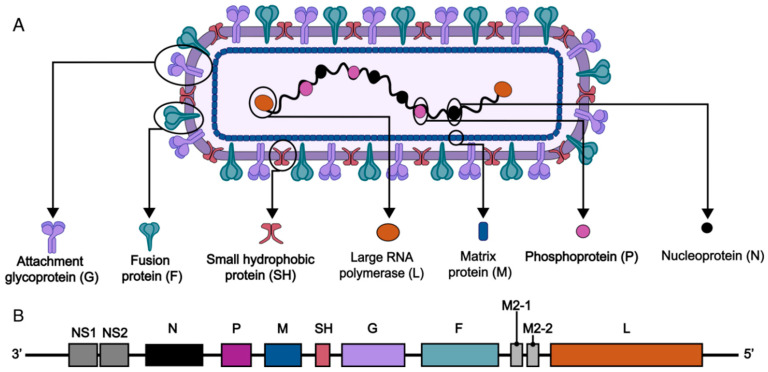
Schematic structure and genome of RSV. (**A**) The virion consists of envelope, matrix and nucleocapsid. The viral envelope is a lipid bilayer containing the fusion (F) and attachment (G) proteins, embedded with small hydrophobic (SH) protein. Beneath the envelope is the matrix protein (M). Inside the envelope, a helical nucleocapsid encapsulates the viral RNA genome, formed by the nucleoprotein (N) and associated with the phosphoprotein (P) and large RNA polymerase (L). (**B**) The viral genome contains genes encoding 11 proteins: the above-mentioned structural proteins F, G, SH, M, N, P and L proteins, and non-structural protein (NS1, NS2) and second matrix protein (M2-1, M2-2).

**Figure 2 vaccines-13-00304-f002:**
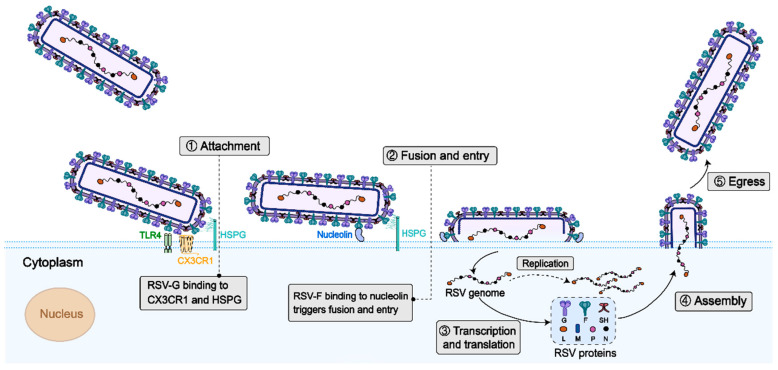
Schematic illustration of RSV life cycle. The lifecycle of RSV initiates with attachment and entry into host cells, mediated by the viral glycoproteins. The G protein engages specific host cell receptors, such as heparan sulphate proteoglycan (HSPG) or CX3CR1, facilitating viral docking. Subsequently, the F protein binds to nucleolin and orchestrates membrane fusion, enabling the release of the ribonucleoprotein (RNP) complex into the cytoplasm, where viral replication and transcription are catalyzed by the RNA-dependent RNA polymerase. Following replication, viral components are transported to the cell membrane, where they assemble into progeny virions. These mature particles, enveloped by a host-derived lipid bilayer, eventually egress from the cell surface.

**Figure 3 vaccines-13-00304-f003:**
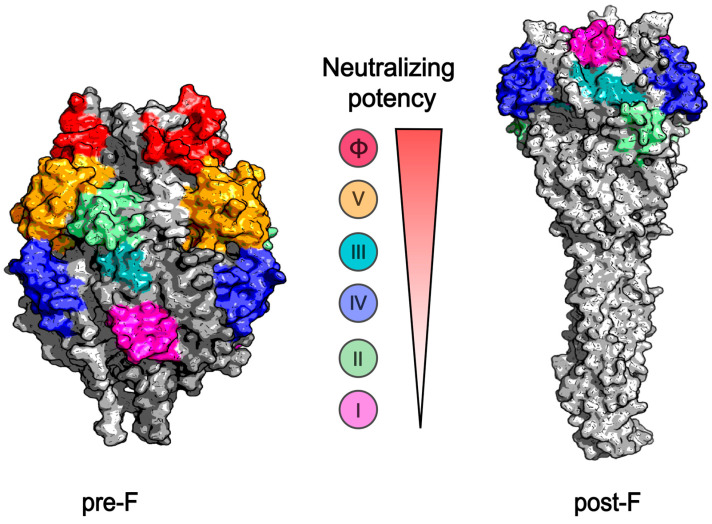
Antigenic sites on the pre-F (**left**) and post-F (**right**) structures of the RSV trimeric F protein, along with the neutralizing capabilities. The surface representation of the protein colored in gray was generated from the Protein Data Bank (PDB) code 4JHW (pre-F) and PDB code 3RKI (post-F), and the antigenic sites labelled with site Ø in red, site V in orange, site III in cyan, site IV in blue, site II in green, and site I in magenta are highlighted on the surface.

**Figure 4 vaccines-13-00304-f004:**
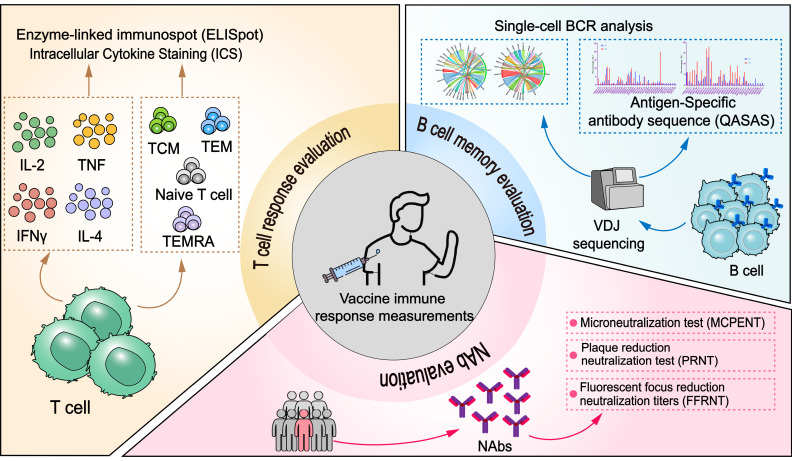
Aspects of vaccine-induced immune response and key technologies applied for vaccine immunological evaluation. Immunogenicity evaluation mainly includes evaluation of humoral, mainly neutralizing, antibodies, and cellular immunity, including the T cell response and B cell evaluation. Neutralizing antibodies (NAbs) are the most important indicator of immunogenicity, and different standardized and high-throughput technologies for NAb measurement have been utilized. T cell responses are usually measured by the quantification of antigen-specific T cells and the presence of different phenotypes of T cells. Memory B cell evaluation is mainly based on single-cell VDJ sequencing. BCR diversity and CDR3 quantification may indicate humoral immune reserve for immunogens, which may set criteria for screening immunogenic candidates.

**Table 2 vaccines-13-00304-t002:** Summary of methods of vaccine evaluation.

Methods	Evaluation Parameters or Indication	Features	Applications
**Neutralizing antibody evaluation**			
MCPENT	Neutralization based on CPE	Based on CPE, needs professional personnel	Preclinical and clinical
PRNT/FRNT	Neutralization based on plaque or foci reduction	Involves staining or immunostaining	Preclinical and clinical
CFFRNT	Neutralizing antibodies based on fluorescence-expressing virus	High-throughput	Preclinical and clinical
**T cell response evaluation**			
ELISpot	Cytokine-positive cells	Easily performed	Preclinical and clinical
ICS	Cytokine-positive cells and the phenotypes, memory phase of T cells	Needs professional personnel, phenotype analysis	Preclinical
AIM	Activation-induced markers	Needs professional personnel, phenotype analysis	Preclinical
Tetramer	Epitope-specific T cells	HLA-restricted	May not be applicable
**B cell evaluation**			
BCR	Germline BCR (VH, VL germlines), BCR repertoire	Involves single-cell BCR technology	Preclinical
QASAS	CDR3 data	Involves single-cell BCR technology	Preclinical

## Data Availability

Data sharing is not applicable to this article as no datasets were generated or analyzed during the current study.

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
