# Peer review of "Progress on Respiratory Syncytial Virus Vaccine Development and Evaluation Methods"

_vaccines, 2025, doi:10.3390/vaccines13030304_

Round 1
Reviewer 1 Report
Comments and Suggestions for Authors
I recommend including, in the second paragraph of the introduction, the information published on Scientific reports 2016 by Killikelly April M et al., about the importance of Pre-fusion F protein conformation as part of hurdles resulting from FI-RSV vaccine.
In line 82, I suggest adding information about the function of F-protein promoting infection between adjacent cells by the syncytia formation.
Line 99-104, Need a bibliographic reference. The authors could cite the information generated by Brian J.Smith and cols in 2002 (Modelling the structure of the fusion protein from human respiratory syncytial virus, Protein Engineering vol. 15). In the Smith’s paper are important information about fusion peptide cleavage site which include potential glycosylation sites. The cleavage site appears to be necessary for activation of the protein and consequently it could be important for the transition to the post-fusion form of the F-protein.
Line 156, I recommend including information about the first attempts at vaccine development in the early 60’s and explaining the important safety issues that arose with its application. It is also relevant to include findings from the study of samples taken from two patients who died during that period. This information has been crucial in improving the development of new vaccines.
Line 168, I consider of importance to establish that the RSV live attenuated vaccine was the second in the development of vaccines against RSV and began in the 80’s. After that brief introduction you could continue with the information already described.
It is necessary to talk about the vaccines that are currently approved. On the one hand, the monoclonal antibody for use in infants and babies which was approved since 2022. On the other hand, Arexvy and Abrysvo were approved in 2023 for use in adults over 60 years of age and pregnant women, respectively.
It is important that the authors update information about the Moderna vaccine (mResvia), which was approved in May 2024.
The authors do not talk about the importance of using the F protein in its pre-F conformation for vaccine development. In addition, they must specify that vector vaccines and attenuated virus vaccines have not been successful. The only successful ones have been the recombinant protein, monoclonal antibody and mRNA.
I recommend deleting the information in section 4.1 as it is not directly relevant to the core topic. Removing this content would allow for a deeper exploration into the topics of section 4.2. Additionally, this would provide the opportunity to offer a more detailed analysis of the four currently approved vaccines, specifying key aspects as efficacy, immunity protection and other relevant data.
I recommend improving the conclusions section. They must conclude that the development of vaccines based on knowledge of the structure and function of the F protein has been very important and has allowed the development of the four vaccines that are currently approved and that appear to be very promising.
Author Response
Comments 1: I recommend including, in the second paragraph of the introduction, the information published on Scientific reports 2016 by Killikelly April M et al., about the importance of Pre-fusion F protein conformation as part of hurdles resulting from FI-RSV vaccine.
Response: We truly thank the reviewer for the kind suggestion. We have cited this paper and added their findings about the F protein on FI-RSV vaccine (line 69–71).
Comments 2: In line 82, I suggest adding information about the function of F-protein promoting infection between adjacent cells by the syncytia formation.
Response: Revised accordingly at line 150–156.
Comments 3: Line 99-104, Need a bibliographic reference. The authors could cite the information generated by Brian J.Smith and cols in 2002 (Modelling the structure of the fusion protein from human respiratory syncytial virus, Protein Engineering vol. 15 ). In the Smith’s paper are important information about fusion peptide cleavage site which include potential glycosylation sites. The cleavage site appears to be necessary for activation of the protein and consequently it could be important for the transition to the post-fusion form of the F-protein.
Response: We appreciate the suggestion. We’ve went through the reference paper mentioned. It demonstrated (may be for the first time) the three-dimensional structure of RSV-F (post-fusion form) with location residues determined for different sites. It is important for the vaccine design that how the findings about post-fusion form have influenced the present theory of structure design. However, as our manuscript overviews the recent progress of RSV vaccines and vaccine evaluations, we think it will be a bit distracted if we elaborate this issue. Thus, we’ve left this unchanged.
Comments 4: Line 156, I recommend including information about the first attempts at vaccine development in the early 60’s and explaining the important safety issues that arose with its application. It is also relevant to include findings from the study of samples taken from two patients who died during that period. This information has been crucial in improving the development of new vaccines.
Response: We appreciate the kind suggestion from the reviewer. Accordingly, we’ve added a few sentences about the first attempts at vaccine development in 1960’s at line 206–213, which have influenced the latter learning about the importance of pre-F conformation.
Comments 5: Line 168, I consider of importance to establish that the RSV live attenuated vaccine was the second in the development of vaccines against RSV and began in the 80’s. After that brief introduction you could continue with the information already described.
Response: We thank the reviewer for the comment. Accordingly, we’ve added the information for better linking to the following line 229–231.
Comments 6: It is necessary to talk about the vaccines that are currently approved. On the one hand, the monoclonal antibody for use in infants and babies which was approved since 2022. On the other hand, Arexvy and Abrysvo were approved in 2023 for use in adults over 60 years of age and pregnant women, respectively.
Response: Thank you for the comments. We’ve mentioned the three approved vaccines in the heading graph of section 3, at line 213–221, with the information about the target population. As we are not focused on monoclonal antibodies, we will omit this part.
Comments 7: It is important that the authors update information about the Moderna vaccine (mResvia), which was approved in May 2024.
Response: Yes, already updated at line 219–221.
Comments 8: The authors do not talk about the importance of using the F protein in its pre-F conformation for vaccine development. In addition, they must specify that vector vaccines and attenuated virus vaccines have not been successful. The only successful ones have been the recombinant protein, monoclonal antibody and mRNA.
Response: We appreciate the reviewer’s suggestion. Indeed it is important information. So we’ve added the information at line 253–254, line 286, and line 325.
Comments 9: I recommend deleting the information in section 4.1 as it is not directly relevant to the core topic. Removing this content would allow for a deeper exploration into the topics of section 4.2. Additionally, this would provide the opportunity to offer a more detailed analysis of the four currently approved vaccines, specifying key aspects as efficacy, immunity protection and other relevant data.
Response: We truly appreciate the reviewer’s comments. When our group started structuring the review, we indeed have discussed about the key aspects of vaccine evaluation. As we aim to discuss the potential and importance of immunological surrogate for future vaccine efficacy. Therefore, we have not put our strength on the part of vaccine safety and efficacy, which already reviewed in some other paper. (1. Zeng B, Liu X, Yang Q, Wang J, Ren Q, Sun F. Efficacy and safety of vaccines to prevent respiratory syncytial virus infection in infants and older adults: A systematic review and meta-analysis. Int J Infect Dis. 2024 Sep;146:107118., 2. Topalidou X, Kalergis AM, Papazisis G. Respiratory Syncytial Virus Vaccines: A Review of the Candidates and the Approved Vaccines. Pathogens. 2023 Oct 19;12(10):1259. , 3. Progress at last against RSV. Nat Med 29, 2143 (2023).). So we’re afraid that we will leave this unchanged.
Comments 10: I recommend improving the conclusions section. They must conclude that the development of vaccines based on knowledge of the structure and function of the F protein has been very important and has allowed the development of the four vaccines that are currently approved and that appear to be very promising.
Response: We truly appreciate the kind suggestions, which will improve the merits of our manuscript. We’ve added the discuss the important fuction of pre-F in neutralizing antibody induction and highlighted the importance of structural design to stabilize pre-F conformation. Seen at line 506–515.
Reviewer 2 Report
Comments and Suggestions for Authors
The authors give a detailed and comprehensive review about vaccine targets, vaccine developments and assessment of efficacy for vaccines against RSV infections. Although RSV is an important respiratory pathogen, the development of an appropriate vaccine proved difficult and large-scale vaccination doesn´t exist. Therefore, a review of the up-to-date status is relevant for the field. The manuscript is well organized and well written.
There are just have a few minor comments:
1. It is interesting, that the authors discuss immunological surrogates for efficacy. In this context – I have just one question for clarification: Since ERD was a problem in early vaccines and this was considered due to certain cytokine responses, does the evaluation of immunological surrogates include tests for the absence of these cytokines? If this is the case, could it be included in the manuscript.
2. L124: Eliminate “While”
3. L29: … with phosphoprotein P, ….
4. L154: Eliminate “However”
5. L222-223: … and a simple …… Is statement correct? Please rephrase the end of this sentence.
6. L281: replace: “ this has not been discussed” by “this is not discussed”
7. L416: Eliminate “However”
Author Response
The authors give a detailed and comprehensive review about vaccine targets, vaccine developments and assessment of efficacy for vaccines against RSV infections. Although RSV is an important respiratory pathogen, the development of an appropriate vaccine proved difficult and large-scale vaccination doesn´t exist. Therefore, a review of the up-to-date status is relevant for the field. The manuscript is well organized and well written.
There are just have a few minor comments:
Comments 1: It is interesting, that the authors discuss immunological surrogates for efficacy. In this context – I have just one question for clarification: Since ERD was a problem in early vaccines and this was considered due to certain cytokine responses, does the evaluation of immunological surrogates include tests for the absence of these cytokines? If this is the case, could it be included in the manuscript.
Response: We highly appreciate the reviewer’s favorable comments about the immunological surrogates for vaccine efficacy. In terms of ERD issue, it is indeed crucial in early vaccines development, which have been proved to the main reason for the failure of FI-RSVs. To the best of our knowledge, ERD is induced due to that FI-RSV elicited almost exclusively binding antibodies, rather than neutralizing antibodies which help block the virus, and elicited CD4+T cells and inflammatory cytokines stead of cytotoxic killer CD8+ T cells. ERD may be examined during the research stage of vaccine development, or may be examined after breakthrough infection. While in terms of immunological surrogates discussed in our manuscript, we mainly focus on the evaluation of the capability of vaccine to induce response that prevent the virus and virus causing disease, and aim to provide information on the identification and validation of immune marker for immunological surrogates. Therefore, ERD may not be included in our manuscript.
Comments 2: L124: Eliminate “While”
Response: We’ve revised and rephrased paragraph.
Comments 3: L29: … with phosphoprotein P, ….
Response: Revised accordingly at line 186.
Comments 4: L154: Eliminate “However”
Response: Revised accordingly at line 202.
Comments 5: L222-223: … and a simple …… Is statement correct? Please rephrase the end of this sentence.
Response: Sorry for the misleading. One word is missing. We’ve revised at line 284–286.
Comments 6: L281: replace: “ this has not been discussed” by “this is not discussed”
Response: Revised accordingly at line 334.
Comments 7: L416: Eliminate “However”
Response: Revised accordingly at line 486.
Reviewer 3 Report
Comments and Suggestions for Authors
This review article (“Progress on respiratory syncytial virus vaccine development and evaluation methods”) by Deng et al. discusses about the recent progresses in human respiratory syncytial virus (RSV) vaccine development and key features of each vaccine technology, focusing on vaccine evaluation, and the feasibility of immunological surrogates for protection. The review article seems to be interesting, is very well-written and quite informative as well. However, I have a few suggestions to the authors before I can recommend it for the publication in the journal Vaccines and I think incorporating them would improve the article. My suggestions are listed below:
1. I believe that this review article can significantly benefit from having a few schematic figures. Having such figures would definitely help to attract a broader audience. These would assist the readers easily understand complex information, such as complex processes, or phenomena that are discussed in the text and also make the article more attractive. In that direction, I have a few suggestions:
(a) An introductory schematic representation of the RSV virion and its genome structure – with proper labelling of each of the segments/components.
(b) Diagram (schematic representation) of RSV life cycle including the cell attachment, fusion and entry.
(c) Steps of fusion; Pre-F and Post-F with the neutralizing epitopes labelled.
(d) Construction/Production of the different types of vaccines. And finally, their delivery routes.
(e) Different immunological evaluations of RSV discussed in the article.
With all these schematic diagrams, I believe that the current discussions in the article would be more meaningful to the readers.
2. The review article can also benefit from having a few summary tables.
3. In the conclusion section, a little more detailed discussion is needed regarding the strengths and drawbacks of each type of the vaccination techniques and neutralizing antibody evaluation methods. This can also be done by including a summary table.
4. Please try to reduce plagiarism as much as possible.
Author Response
- I believe that this review article can significantly benefit from having a few schematic figures. Having such figures would definitely help to attract a broader audience. These would assist the readers easily understand complex information, such as complex processes, or phenomena that are discussed in the text and also make the article more attractive. In that direction, I have a few suggestions:
Comments 1: (a) An introductory schematic representation of the RSV virion and its genome structure – with proper labelling of each of the segments/components.
(b) Diagram (schematic representation) of RSV life cycle including the cell attachment, fusion and entry.
(c) Steps of fusion; Pre-F and Post-F with the neutralizing epitopes labelled.
(d) Construction/Production of the different types of vaccines. And finally, their delivery routes.
(e) Different immunological evaluations of RSV discussed in the article.
With all these schematic diagrams, I believe that the current discussions in the article would be more meaningful to the readers.
Response: We are grateful to the reviewer for the thoughtful and valuable suggestions. We’ve generated four figures (Figure 1-4) for a, b c, and e suggestions, and they indeed help to refine our manuscript. However, we’ve left the suggestion d unresolved, as vaccine types towards different delivery routes may be too complex and if we elaborate the issues, it may be a little bit distracted.
Comments 2: The review article can also benefit from having a few summary tables.
Response: We thank the reviewer for the suggestion. Accordingly, two tables, one on the summary of vaccines reviewed in the paper, and one on the summary of vaccine evaluation methods have been created and added to our manuscript.
Comments 3:. In the conclusion section, a little more detailed discussion is needed regarding the strengths and drawbacks of each type of the vaccination techniques and neutralizing antibody evaluation methods. This can also be done by including a summary table.
We appreciate the reviewer for the kind suggestion. We’ve revised through the whole section in the conclusions carefully according to the suggestions, which have strengthen our work.
Comments 3: Please try to reduce plagiarism as much as possible.
Response: We sincerely thank the reviewer for pointing this problem out. We are sorry if this has brought any confusion or inconvenience. We’ve looked carefully and revised the whole manuscript thoroughly to avoid the problem.
Round 2
Reviewer 3 Report
Comments and Suggestions for Authors
The authors have made a sincere effort to answer most of my comments and queries. They have performed quite a bit of additional writing and modifications with appropriate literature support in response to the raised concerns. As a result, the quality of the review article after the revision, has improved significantly. I, therefore, strongly recommend the acceptance of the article for publication in the Vaccines journal.